Method

# Detecting sequence signals in targeting peptides using deep learning

Jose Juan Almagro Armenteros[1,*], Marco Salvatore[2,3,*] , Olof Emanuelsson[2,4] , Ole Winther[5,6,7], Gunnar von Heijne[2,3] , Arne Elofsson[2,3] , Henrik Nielsen[1]

In bioinformatics, machine learning methods have been used to predict features embedded in the sequences. In contrast to what is generally assumed, machine learning approaches can also provide new insights into the underlying biology. Here, we demonstrate this by presenting TargetP 2.0, a novel state-of-the-art method to identify N-terminal sorting signals, which direct proteins to the secretory pathway, mitochondria, and chloroplasts or other plastids. By examining the strongest signals from the attention layer in the network, we find that the second residue in the protein, that is, the one following the initial methionine, has a strong influence on the classification. We observe that two-thirds of chloroplast and thylakoid transit peptides have an alanine in position 2, compared with 20% in other plant proteins. We also note that in fungi and single-celled eukaryotes, less than 30% of the targeting peptides have an amino acid that allows the removal of the N-terminal methionine compared with 60% for the proteins without targeting peptide. The importance of this feature for predictions has not been highlighted before.

## Introduction

The localisation of proteins in the cell is a fundamental determinant of protein function. Specific sorting signals drive the subcellular localisation of proteins. These signals vary in structure, length, and position between the different subcellular compartments. One of the most common types of sorting signals are the N-terminal targeting peptides. These signals are responsible for sorting proteins to the secretory pathway, mitochondria, chloroplasts (or other plastids), and compartments inside the chloroplast such as thylakoids.

Signal peptides (*SPs*) are responsible for transporting proteins to the endoplasmic reticulum to enter the secretory pathway. *SPs* are composed of three regions: a positively charged domain or n-region, a hydrophobic core or h-region, and a segment before the cleavage site (CS) or c-region (von Heijne, 1990).

Mitochondrial transit peptides (*mTPs*) are responsible for targeting proteins to the mitochondrial matrix. *mTPs* are usually enriched in arginine, leucine, and serine. Moreover, they tend to form an amphiphilic helical structure to interact with the import receptor on the mitochondrial membrane (von Heijne, 1986). Proteins targeted to the inner mitochondrial membrane or the inter-membrane space often have a bipartite *mTP*, where the second part is similar to an *SP* (Stuart & Neupert, 1996) (Fig S1).

Chloroplast transit peptides (*cTPs*) are involved in the transport of proteins to the chloroplast stroma. Most of the *cTPs* consist of three regions: an uncharged N-terminal region, a central region lacking acidic amino acids but enriched in serine and threonine, and a C-terminal region enriched in arginine that forms an amphiphilic $\beta$ strand (Heijne et al, 1989a). In addition, chloroplastic proteins targeted to the thylakoid lumen have a bipartite pre-sequence structure (Robinson & Klsgen, 1994). Once the *cTP* is cleaved and the protein enters the stroma, a luminal transit peptide (*luTP*) is recognised, and the protein is further transported to the thylakoid, where the *luTP* is cleaved. The *luTP* is similar to a bacterial *SP*, and the thylakoidal processing peptidase belongs to the family of signal peptidases (Packer and Howe, 2013).

As these signals direct the transport of proteins within the cell, it is crucial to be able to predict their presence in protein sequences accurately. For this reason, in the last two decades, many tools have been developed. Those adopt various machine learning algorithms, including Grammatical Restrained Hidden Conditional Random Fields, N-to-1 Extreme Learning Machines, Support Vector Machines, Markov chains, profile-hidden Markov models, and neural networks (Emanuelsson et al, 2000; Small et al, 2004; Petsalaki et al, 2006; Fukasawa et al, 2015; Savojardo et al, 2015).

One of the most used methods is TargetP 1.1 (Emanuelsson et al, 2000). TargetP uses feed-forward networks and position-weight matrices

[1]Department of Health Technology, Section for Bioinformatics, Technical University of Denmark, Kongen Lyngby, Denmark    [2]Science for Life Laboratory, Solna, Sweden    [3]Department of Biochemistry and Biophysics, Stockholm University, Stockholm, Sweden    [4]Department of Gene Technology, School of Engineering Sciences in Biotechnology, Chemistry and Health, KTH—Royal Institute of Technology, Stockholm, Sweden    [5]DTU Compute, Technical University of Denmark, Kongen Lyngby, Denmark    [6]Computational and RNA Biology, University of Copenhagen, Copenhagen, Denmark    [7]Centre for Genomic Medicine, Rigshospitalet, Copenhagen University Hospital, Copenhagen, Denmark

Correspondence: arne@bioinfo.se; henni@dtu.dk
*Jose Juan Almagro Armenteros and Marco Salvatore contributed equally to this work

to process windows of amino acids to predict the presence of *SPs*, *mTPs*, and *cTPs* and the positions of their CSs. However, with the rise of deep learning, new types of networks such as recurrent neural networks (RNNs) have gained popularity. The main reason is their extraordinary ability to work with sequence data and model long-range relationships between inputs in the sequence.

RNNs sequentially process sequences of any length, being able to retain information from previous positions in the sequence. Several methods have taken advantage of this type of network to try to better predict signal and transit peptides (Reczko & Hatzigeorgiou, 2004; Boden & Hawkins, 2005). These methods make use of bidirectional RNNs (BiRNN), which are two RNNs, one processing the sequence forwards and another processing the sequence backwards. With this construction, the context around each amino acid is modelled, as the forward RNN processes all the amino acids from the N terminus up to one position and the backward RNN processes all the amino acids from the C terminus up to the same position.

However, regular RNNs, the so-called Elman networks, are challenging to train (the so-called exploding/vanishing gradient problem) and often fail to capture dependencies far apart in the sequence (Pascanu et al, 2012 *Preprint*). Therefore, the ability of the network to hold information from multiple steps back is reduced. A variant of the RNN cell, the long short-term memory (LSTM), solves this problem by a construction akin to a computer memory cell that holds information for multiple steps. This type of RNN cell together with BiRNN has been successfully applied to the prediction of *SPs* and *mTPs* (Thireou & Reczko, 2007; Almagro Armenteros et al, 2019). Today, new methods, such as DeepLoc (Almagro Armenteros et al, 2017), use bidirectional LSTM (BiLSTM) to predict the localisation of proteins to a broader range of compartments. DeepLoc accurately predicts the localisation of proteins but not the presence of the N-terminal sorting signals and the position of the CSs. Starting from this architecture, we decided to develop TargetP 2.0 using BiLSTM and a multi-attention mechanism. Using the multi-attention mechanism, the network can predict both the type of peptide and the position of the CS by focusing on particular regions of the sequence.

Moreover, we assemble a new protein dataset that we use to train TargetP 2.0 (http://www.cbs.dtu.dk/services/TargetP-2.0/). TargetP 2.0 can jointly predict the presence of SPs, mitochondrial, chloroplast and thylakoid transit peptides, and the corresponding CS positions. TargetP 2.0 is available at http://www.cbs.dtu.dk/services/TargetP-2.0/, and the source code is available under the creative commons CC BY-NC-SA license from https://github.com/JJAlmagro/TargetP-2.0/.

When analysing the attention layer from the final version of the network, it became apparent that most information was retrieved from two distinct positions in most sequences. One of these was, as expected, localised close to the CS. However, an equally important signal from position 2 in the sequences was also found. Next, we examined the amino acid frequencies in the second position, after the first methionine, of all proteins. To our surprise, very distinct patterns emerged. In chloroplasts and plastids, the second residue was frequently an alanine, whereas in all targeting peptides in fungi and unicellular eukaryotes amino acids that allow cleavage of the methionine are rare (see Fig 1).

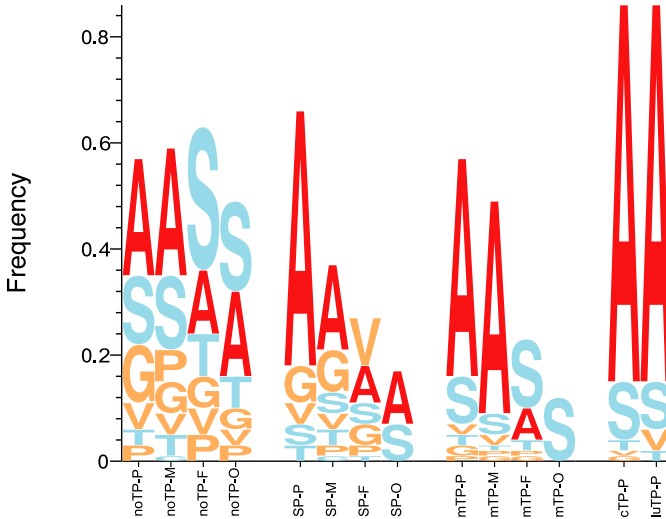

**Figure 1. This figure depicts the frequencies of the second residue in proteins with different targeting peptides.**
The proteins are divided into their respective type of targeting peptide: signal peptide (*SP*), mitochondrial transit peptides (*mTPs*), chloroplast transit peptides (*cTPs*), luminal transit peptides (*luTPs*), and *noTPs*. Furthermore, the proteins were divided into their kingdom: Viridiplantae (P), Metazoa (M), Fungi (F), and other eukaryotic organisms (O) sequences. Inspired by sequence LOGOs, the height of each letter corresponds to the frequency of that amino acid. Only the frequencies for the short side-chained amino acids that allow the cleavage of the N-terminal methionine are shown.

# Results and Discussion

Here, we have developed a deep learning model to predict targeting peptides described in Fig 2. We compare TargetP 2.0 with state-of-the-art predictors on a set of proteins with experimentally verified targeting peptides.

### TargetP 2.0 improves identification of targeting peptides

In Table S1, it can be seen that TargetP 2.0 is better than all the competitors at the identification of targeting peptides in accuracy and correlation coefficients. From the receiver operator curves in Fig 3, it is clear that TargetP 2.0 performs better than the alternative methods except SignalP 5.0 for *SPs* for identification of all four targeting peptides. It can also be noted that the identification of SPs is more reliable than the identification of transit peptides. TargetP 2.0 predicts ~97% of the *SPs* correctly compared with less than 90% for other targeting peptides (see Table S1). For non-plant proteins, the most common confusion is between *mTPs* and non-TPs (see Table S2).

The poor discrimination between *mTP* and *cTP* of TargetP 1.1 and other older methods has been significantly improved in TargetP 2.0. The number of correctly predicted peptides increased from about 50 to 90%. The only other method that shows a similar performance is DeepLoc, which is based on a similar methodology and training set but cannot predict CSs. TargetP 2.0 also performs significantly better at the identification of *cTPs* and *luTPs* than PredSL (Petsalaki et al, 2006), the only other method that can identify *luTPs*. However, still 11 of 45 *luTPs* are classified as *cTPs* (see Table S3).

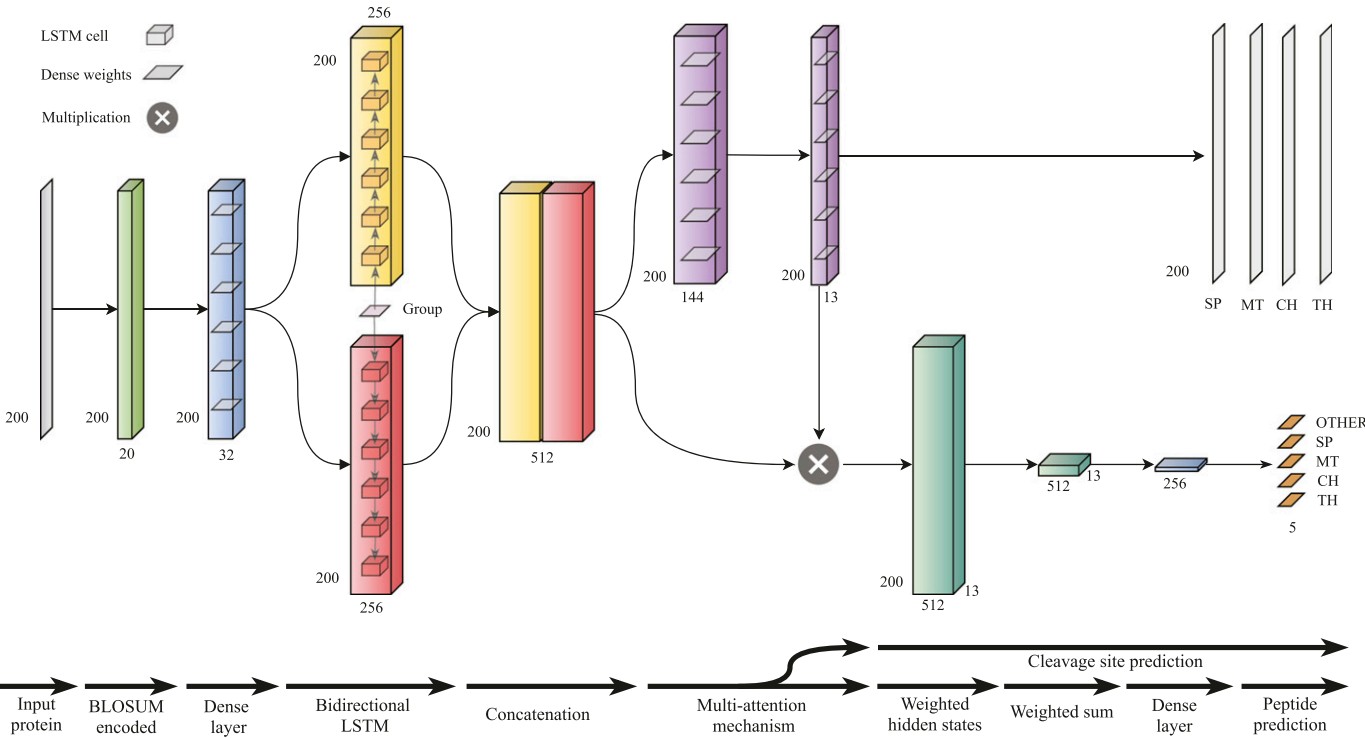

**Figure 2.  The TargetP 2.0 architecture.**

It can be seen that a very simple method that only considers the 20 N-terminal amino acids, MLP-20, performs on par with previous methods when it comes to *mTPs* and *SPs*, but slightly worse than Predotar for *cTPs*. Even when only using 10 residues, MLP-10 performs better than PredSL for all categories except *luTPs* (see Table S1).

A more detailed analysis at the kingdom level for TargetP 2.0 can be found in Table S4. Here, we can see that the prediction accuracy is slightly lower in Fungi than in the other kingdoms. One possible explanation could be that the GC content in the Fungi genomes is lower than that in the other genomes. The low GC content affects the amino acid frequencies, making alanine less frequent (Basile et al, 2017).

Because the chloroplast is not the only type of plastid, we finally tested the ability of TargetP 2.0 to predict proteins of amyloplasts and chromoplasts, which differ from chloroplasts primarily through their pigments. UniProt provides transit peptide annotation for 10 amyloplast and 32 chromoplast proteins. TargetP 2.0 predicts 9 of 10 amyloplast and 26 of 32 chromoplast proteins to have a *cTP*, achieving a similar performance for these plastid proteins.

### TargetP 2.0 improves the prediction of CSs in *cTPs* and *luTPs*

We tested the CS prediction ability on the test set and only for the correctly predicted proteins. The CS prediction is best for *SPs*, with a recall (accuracy) of 83% on the test set both for TargetP 1.1 and TargetP 2.0 (see Fig 4) and Table S5. TargetP 2.0 does not reach the accuracy obtained by SignalP 5.0 (86%) possibly because of the use of a conditional random field to predict the CS in SignalP 5.0 or

because many of the proteins we tested are included in the training set of SignalP 5.0. Anyhow, when allowing a positional shift of up to five residues, the performance is identical (96%).

In *mTP* and *cTP* CS, prediction is more difficult with a recall of 46 and 49% by TargetP 2.0, respectively. However, this is a clear improvement over TargetP 1.1 and all other methods for *cTPs*, and a slight improvement for *mTPs*.

TargetP 2.0 CS predictions of the *luTP* is a new feature. Given the small number of peptides in the database, the recall of 60% (27 correctly identified *luTP* CSs) is better than expected and a significant improvement over the only other method that can predict *luTPs*, PredSL (Petsalaki et al, 2006), which only identifies 5 (11%) CSs correctly.

If we allow up to five residue shifts of the prediction of CSs, about two-thirds of the CSs in *cTPs*, *mTPs*, and *luTPs* can be identified correctly (see Table S5).

### Comparison with UniProt annotations

TargetP 2.0 provides a possibility for fast and accurate annotation of entire or incomplete proteomes in a few hours, as it takes on average only 0.20 s to run a single protein on a dedicated 8-core machine. We annotated several eukaryotic proteomes for a total of 288,964 proteins from six Metazoa (*Caenorhabditis elegans*, *Drosophila melanogaster*, *Danio rerio*, *Homo sapiens*, *Mus musculus*, and *Xenopus tropicalis*), five Viridiplantae (*Arabidopsis thaliana*, *Brachypodium distachyon*, *Oryza sativa*, *Solanum lycopersicum*, and *Vitis vinifera*), and two Fungi (*Saccharomyces cerevisiae* and *Schizosaccharomyces pombe*) proteomes. All predictions are available from the accompanying website.

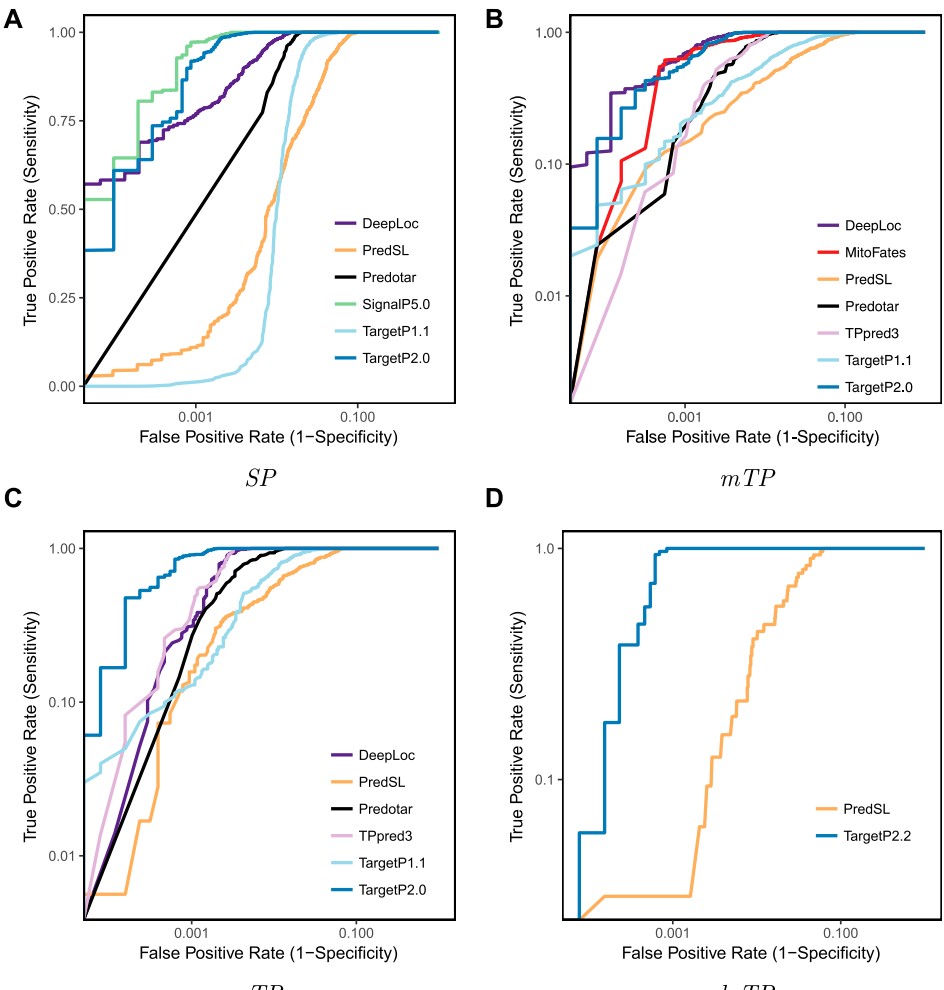

Figure 3. Receiver operator curves for identification of SPs, mitochondrial-, chloroplast-, and luminal transit peptides.

We examined the possibility to modify the number of annotated proteins using the confusion matrix of TargetP 2.0 as proposed before (Marot-Lassauzaie et al, 2018). In this method, the predicted number of proteins in a class is calculated by using the number of members predicted to that class and the estimated number of mispredictions from other classes. First, the fraction of misprediction between classes is calculated. The estimated number of members of one class is then calculated from the predicted number in that class, deducing the estimated fraction of false predictions. Thereafter, the estimated number of mispredictions from other classes is added from the number of predicted members in a class multiplied with the estimated mispredictions from that class to the first class. The number of peptides in each class changed with less than 3% for all categories except the *luTPs* that were under-predicted by 25% (see Table S6). This indicates that our estimates of the number of targeting peptides of each type should be rather accurate except for *luTPs*.

In Table S7, a comparison of the annotations from TargetP 2.0 and UniProt is presented, including also the proteins annotated by electronic annotations, that is, predictions. These annotations on UniProt are certainly of lower quality than the ones used in our dataset. Therefore, this comparison should not be taken as a measurement of the performance of a particular prediction tool. Instead, they should be seen as an estimate of how well we can annotate genomes automatically today. For the best annotated proteomes, *H. sapiens*, *M. musculus*, *S. cerevisiae*, and *A. thaliana*, the agreement between UniProt and TargetP 2.0 predictions is about 80% for the organelles and more than 90% for SPs. The high agreement for *SPs* is quite likely due to UniProt applying SignalP (Petersen et al, 2011) for its annotation of *SPs*, and it was trained on a similar dataset as used here. For the other proteomes, the agreement is substantially worse, except for *SPs*, indicating that the transit peptide annotation in UniProt is far less complete than the *SP* annotation and that applying TargetP 2.0 would significantly improve the annotation.

A few interesting differences can be observed, which might have biological relevance. TargetP 2.0 predicts about twice as many mitochondrial proteins in plant proteomes compared with metazoan proteomes. Even in *A. thaliana*, only half of these proteins are annotated in UniProt as mitochondrial. In agreement with the UniProt annotations, fungi seem to have fewer mitochondrial proteins than other eukaryotes. The number of predicted chloroplast proteins varies significantly between the proteomes, from 1,125 in the grape proteome to 2,049 in the rice proteome. However,

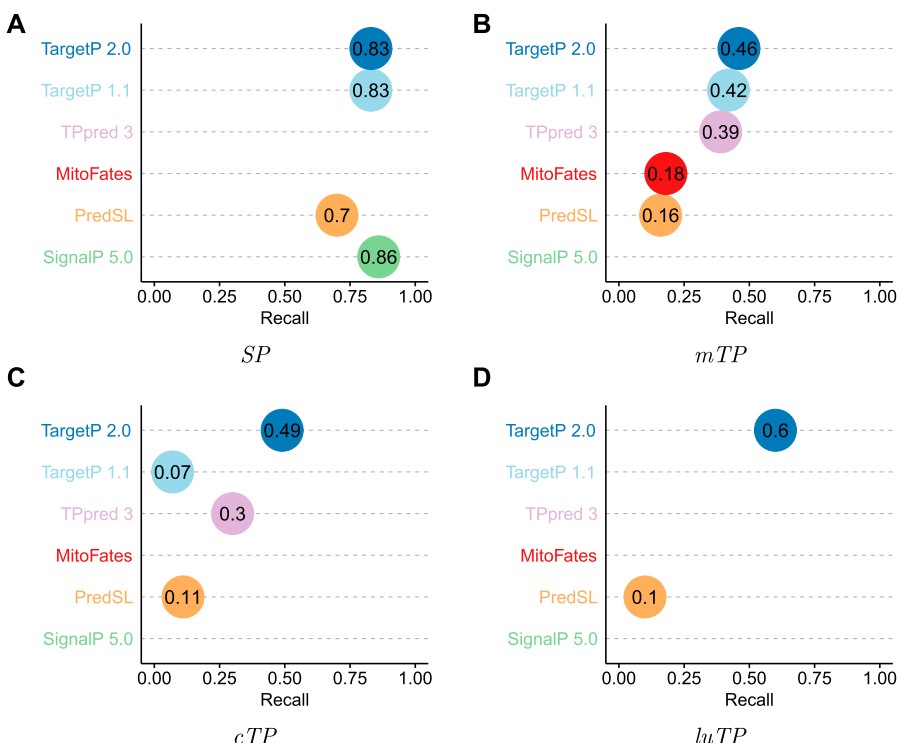

Figure 4. Recall (or accuracy) for the CS prediction in *SP*s, *mTP*s, *cTP*s, and *luTP*s by the different prediction methods. Note that not all methods can predict all types of targeting peptides.

### Identification of the strongest contributing sequence factors

In the previous paragraphs, we show that by using a deep learning architecture, it is possible to improve the prediction of targeting peptides. Next, we wanted to examine if it is possible to extract which biological features contribute to improved performance.

To analyse which features the deep learning model learned, we focused on the maximum outputs from the attention layer (see Fig S1). It is clear that for most proteins with targeting peptides, there are two positions with strong signals, one very close to the N terminus (at position 2) and one later corresponding to a position close to the CS. These positions were analysed in more detail, by aligning all the proteins either starting from the predicted CS (Fig 5) or from the N terminus (Fig 6).

In Fig 5, it can be seen that the attention layer focuses on the position just before the CS (the −1 position). In *SP*s, *cTP*s and *luTP*s position −1 is dominated by alanine, whereas in *mTP*s, this position is dominated by tyrosine and phenylalanine. In addition, the actual cleavage signal comes from a couple of positions (such as −1 and −3 in *SP*s and *luTP*s) and not only position −1 (see Figs 7 and S2). This difference can be explained by the attention layer collapsing information from nearby positions into one position and not the biological importance of position −1. In addition to the site close to the CS, most of the information obtained from the attention layers is directly the N terminus of the CS. In agreement with what is known about the differences between the targeting peptides, the attention for the *SP*s is focused on a stretch of ~10 hydrophobic

residues, whereas the other peptides have a longer stretch of informative residues. As is well known, the *mTP*s are enriched in arginine.

### TargetP 2.0 overpredicts *mTP* CSs with arginine in −3

For the *SP*s, *cTP*s, and *luTP*s, the sequence logos are almost identical between predicted and experimentally annotated proteins, both in the CS and the signal composition. However, we can observe that for *mTP*s, the amino acid composition near the CS differs between predicted (Fig 7B) and experimentally verified *mTP*s (Fig S2). In both cases, there is an abundance of arginines in position −2, −3, and −10 from the CS as described before (Gavel & von Heijne, 1990; Kutejov et al, 2013). However, the signal for arginine at −3 is stronger among the predicted than among the experimentally verified CSs. To investigate this difference further, we plotted the distribution of the distance from the experimental and predicted CSs to the nearest upstream arginine (see Fig S3). It shows that although there is good agreement at most positions, there is a clear overprediction at −3 and an underprediction at −10.

The sites with arginine at −2 are thought to represent the original cleavage by mitochondrial processing peptidase, whereas the sites with arginine at −3 and −10 are thought to arise by subsequent cleavage events by the Icp55 peptidase and mitochondrial intermediate peptidase, respectively (Vögtle et al, 2009; Gakh & Isaya, 2013; Kutejov et al, 2013; Vögtle & Meisinger, 2013; Fukasawa et al, 2015). The cleavage by Icp55 could explain the fact that some patterns in the *mTP* CS (Figs 7B and S2B) seem to be repeated with a shift of one position, for example, the preference for serine that occurs in positions 1 and 2 in the mature protein.

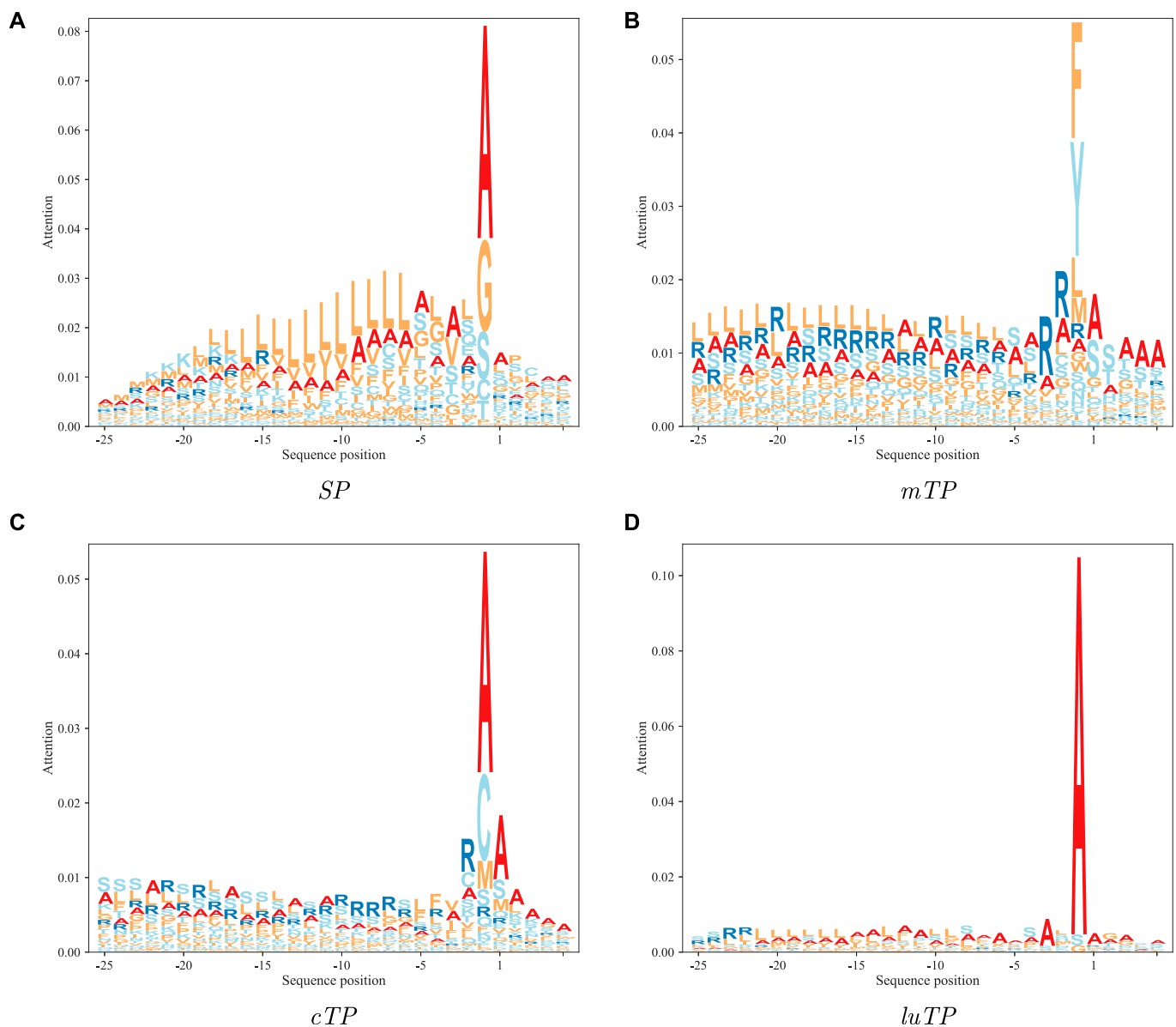

**Figure 5.** Attention layer LOGOs showing the impact strength of the attention layer and the frequency of amino acids. All sequences are aligned at the predicted CS.

The findings represented in Fig S3 show that the model can easily recognise the arginines at position −2 (original Mitochondrial Processing Peptidase sites) and −3 (Icp55 sites), but has troubles in identifying arginines at position −10 (mitochondrial intermediate peptidase sites). This overrepresentation of arginine at position −3 and underrepresentation at position −10 is probably contributing to the relatively low performance on the CS prediction in *mTPs*. It might be relevant to explore further the distance of arginines from the CS and the patterns recognised by the three peptidases to improve the prediction of the *mTP* CS in future versions.

### *cTPs* have an alanine in position 2

There is also a strong attention peak at position 2 for all targeting peptides (see Fig 6). From the sequence logo, it is clear that position

2 amino acid preferences differ between targeting peptides (see Figs 8 and S4). In *cTPs* and *luTPs*, there is a powerful signal for alanine in position 2. In contrast, SPs have some preference for lysine and *mTPs* for alanine or leucine in position 2 (see Table S8).

The importance of position 2 is likely to be related to the cleavage of the N-terminal methionine. When there is a short side-chained amino acid (Ala, Cys, Gly, Pro, or Ser) in position 2, the methionine can be cleaved by a methionine aminopeptidase (MAP) (Frottin et al, 2006). There exist two classes of MAPs, MAP1, and MAP2. All these proteins are homologous to the machinery in bacteria, indicating that they work co-translationally. *A. thaliana* has four MAP1s (MAP1A, MAP1B, MAP1C, and MAP1D) and two MAP2s (MAP2A and MAP2B). It has been shown that MAP1B, MAP1C, and MAP1D are targeted for proteins belonging to the organelles (Giglione & Meinnel, 2001).

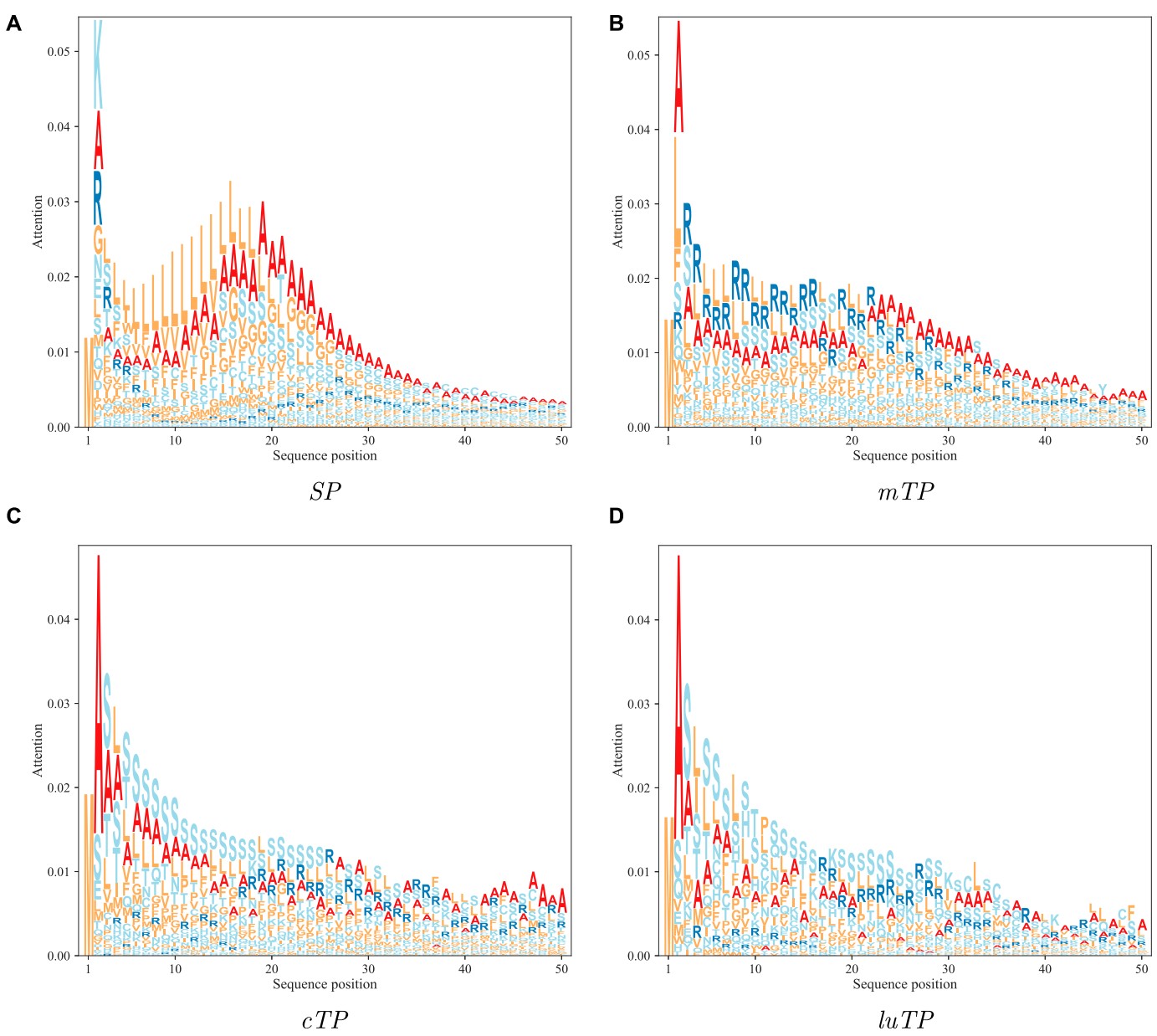

**Figure 6. Attention layer LOGOs showing the impact strength of the attention layer and the frequency of amino acids. All sequences are aligned at the N terminus.**

In Fig 1, it can be seen that about 60% of the proteins without targeting peptides (*noTPs*) have an amino acid in position 2 that allows the N-methionine to be cleaved. These proteins have mostly alanine or serine in position 2. The N-terminal methionine can only be cleaved if the second residue has a short side chain. For proteins with SPs, in all species except the plants, less than 40% of the residues in position 2 have a short side chain. The same can be seen for *mTPs* in the fungi and single-celled eukaryotic groups.

Most striking is the observation that about two-thirds of the *cTPs* and *luTPs* have an alanine in position 2 (see Fig 1). This preference has been noted before (Heijne et al, 1989b; Zybailov et al, 2008). When mutating the second position in dual-targeting proteins that are imported to both chloroplasts and mitochondria, the targeting was disrupted (Pujol et al, 2007). Surprisingly, when the authors

mutated one of the few chloroplast proteins that did not have an alanine in position 2, *PheRS*, from threonine to alanine the import to chloroplasts decreased.

It has been reported that amino acid frequencies in position 2 differ between species (Shemesh et al, 2010). The frequency of alanine in position 2 varies from 7% in *Escherichia coli* to close to 30% in *A. thaliana*. Table S9 shows that alanine is frequent in all types of proteins in *A. thaliana* but also that the frequency is higher in proteins targeted for plastids. One possible reason for alanine to be preferred in position 2 is that alanine has a strong helical propensity. The amino-terminal sections of *cTPs* and *luTPs* are less prone to form secondary structures than *mTPs* and *SPs* at the amino-terminal (see Fig S5). Here, it can also be seen that SPs have a stronger tendency to form a structure close to the N terminus

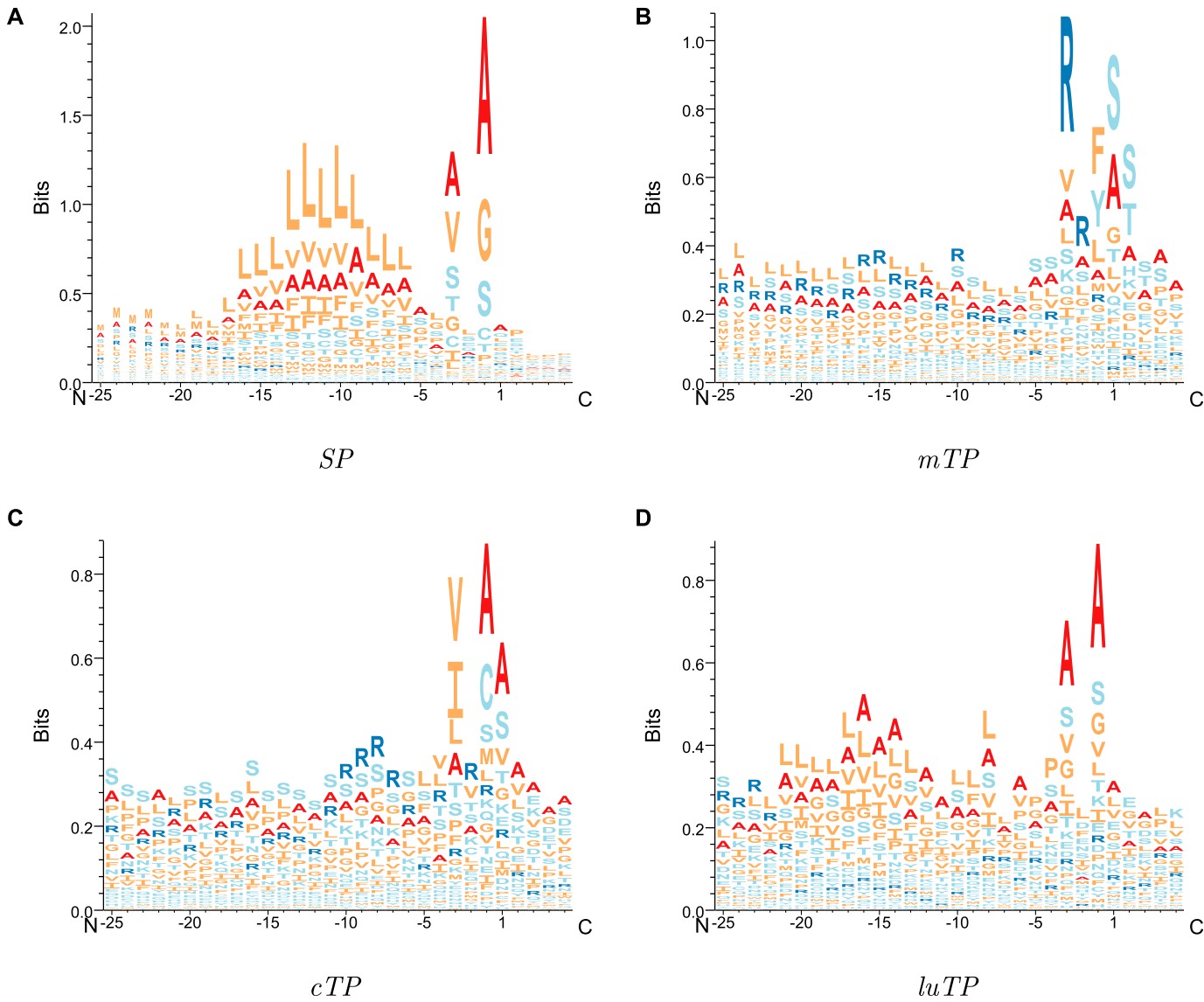

**Figure 7. Sequence LOGOs showing the amino acid frequencies in the pre-sequences.**
All sequences are aligned according to the predicted CS.

than the other peptides. The importance of the N termini can also be seen by the fact that the simple MLP-20 method performs quite well at identification of *noTPs*, *SPs*, and *mTPs*. However, to fully understand the importance of the second position, additional experimental studies are needed.

## Materials and Methods

### Dataset

The protein data used to train TargetP 2.0 were extracted from the UniProt database, release 2018_04 (UniProt-Consortium, 2014). The negative dataset consists of proteins without either signal or transit peptides from the nucleus, cytoplasm, and plasma membrane

(without SPs) and with experimental annotation (ECO:0000269) of their subcellular localisation. The positive set contained secreted, mitochondrial, chloroplastic, and luminal proteins with experimental annotation of their signal or transit peptide. The final set consists of 9,537 *noTPs*, 2,697 with *SPs*, 499 *mTPs*, 227 *cTPs*, and 45 *luTPs* (see Table S1). Note that although a thylakoid targeting signal, as described in the introduction, consists of a *cTP* followed by an *SP*-like *luTP*, the first CS (for the stromal processing peptidase) is almost never annotated in UniProt. We were, therefore, not able to predict this CS for thylakoid proteins, only the second cleavage by thylakoidal processing peptidase will be predicted. Hereafter, "*luTP*" will refer to the entire thylakoid targeting signal. The dataset was further divided into four groups representing the eukaryotic kingdoms Viridiplantae, Metazoa, and Fungi and a group of other eukaryotes.

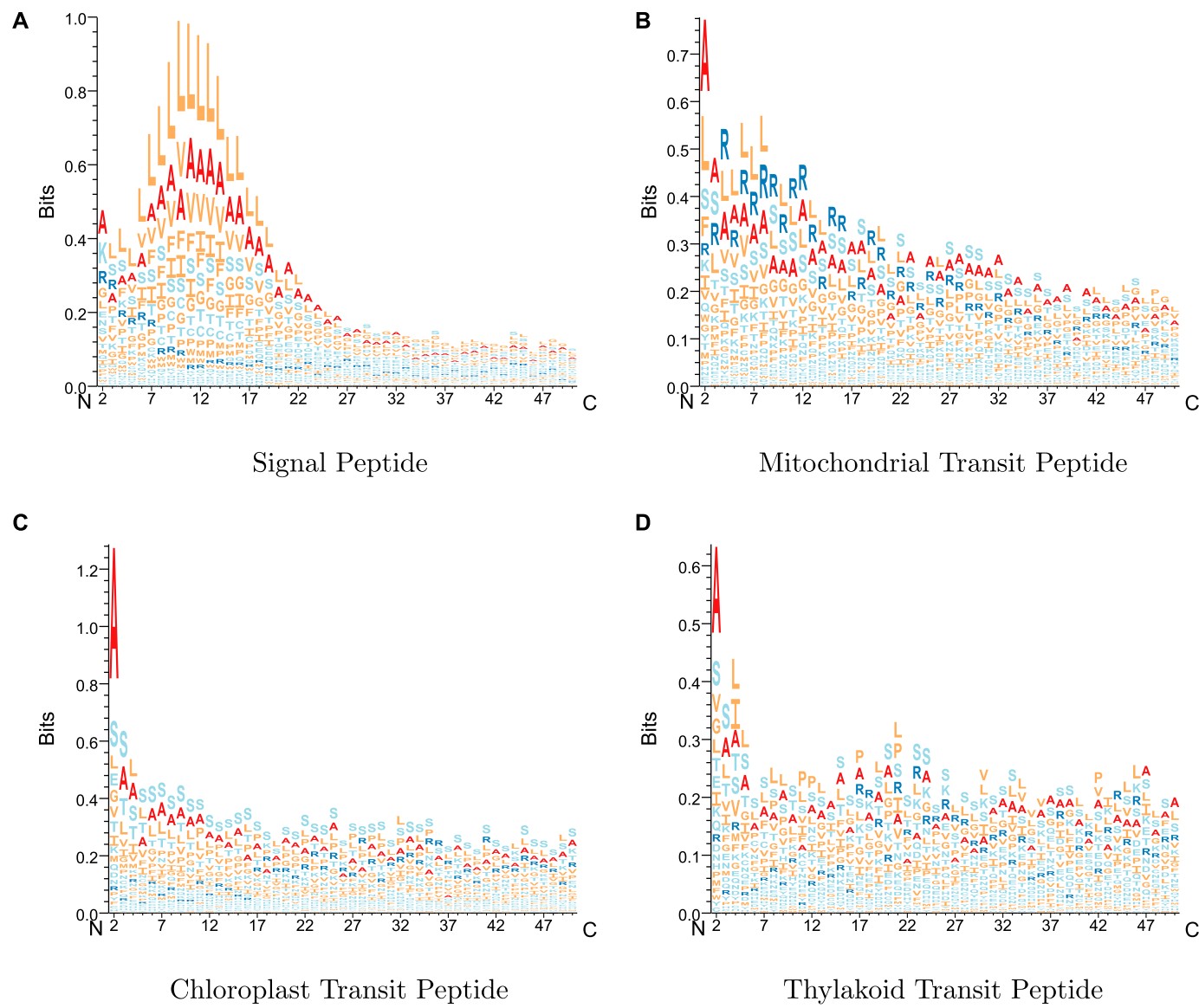

**Figure 8. Sequence LOGOs showing the amino-terminal pre-sequences.**
All sequences are aligned at the N terminus.

We also trained unique predictors for each eukaryotic kingdom; unfortunately, this resulted in a decrease in performance by about 5%. Most likely, this is due to smaller training sets and that the targeting peptides do not differ significantly between the kingdoms.

PSI-CD-HIT (Li & Godzik, 2006) was used to cluster the first 200 residues of each protein with 20% of identity or $10^{-6}$ E-value using Basic Local Alignmst Search Tool and alignment coverage of at least 80% of the shorter sequence. We performed a stringent homology partitioning to get a realistic assessment of generalisation performance. Each cluster of homologous proteins was assigned to one of five cross-validation groups to ensure that similar proteins were not mixed between the different datasets.

### The TargetP 2.0 algorithm

The TargetP 2.0 model is described in Fig 2. The model consists of two key components, a BiRNN with LSTM cells and a multi-attention mechanism (Lin et al, 2017 *Preprint*) to predict both the type of peptide and the position of the CS.

The input to this model is the first 200 amino acids of a protein. This threshold was chosen based on the maximum length of known transit peptides, which is 162 amino acids (Stefely et al, 2015). The amino acids in the protein are encoded using BLOSUM62 substitution matrices.

We first describe the model at a high level and give more details on each of the layers below: The first layer of the model is a fully connected layer to perform a feature transformation of each amino acid input feature with 32 hidden units. The following layer is the BiLSTM with 256 hidden units in both forward and backward directions. The first hidden state to the BiLSTM is a vector containing the group information, which denotes whether the protein is a plant or nonplant protein. The 512-dimensional concatenated output from

the BiLSTM is then used to calculate the multi-attention matrix similarly to those applied in machine translation (Bahdanau et al, 2014 *Preprint*; Luong et al, 2015 *Preprint*). The attention size is 144 units and the number of outputs from the attention matrix is of size 13. Of these 13 attention vectors, four were used to predict the different CS positions for *SP*, *mTP*, *cTP*, and *luTP*. The attention matrix is further utilised to encode the whole sequence into a context matrix. This context matrix of size 512 × 13 is processed by a fully connected layer with 256 units, to summarise it into a vector. Finally, this is fed to the output layer with 5 units and softmax activation.

We train a model that learns to predict the type of peptide and the position of the corresponding CS ($y$, $y'$) where $y$ is the predicted type of peptide, $y'$ the predicted CS position, $f$ the model, $\theta$ the learnable parameters, and $X$ the protein sequence. Here, $y$ is a vector of size equal to the number of classes $C$, five in this case, and $y'$ is a vector of size equal to the length of the sequence $L$, which can be up to 200. The $\theta$ parameters are optimised using an extension of stochastic gradient descent, Adam with cross-entropy loss for both types of peptide and CS prediction. Both losses were then averaged. The only regularisation technique used was dropout between the different layers.

The network has three main types of layers: fully connected, RNN with LSTM cell, and multi-attention layer. The first fully connected layer $c$ applies a feature transformation:

$$c_t = f_c(Wx_t + b) \tag{1}$$

where $x_t$ is an amino acid at position $t$ in the sequence and $W$ and $b$ are the learnable weights and biases. The first layer is followed by a BiRNN that utilises an LSTM cell to capture the context around each amino acid in the sequence. The RNN applies the same set of weights to each position $t$

$$\overrightarrow{h_t} = \overrightarrow{LSTM}\left(c_t, \overrightarrow{h_{t-1}}\right) \tag{2}$$

$$\overleftarrow{h_t} = \overleftarrow{LSTM}\left(c_t, \overleftarrow{h_{t+1}}\right) \tag{3}$$

where $\overrightarrow{h_t}$ and $\overleftarrow{h_t}$ are the hidden states of the RNN at position $t$ for the forward and backward directions, respectively. The hidden states are concatenated into $[\overrightarrow{h_t} ; \overleftarrow{h_{\downarrow t}}]$.

The last part of the network is a multi-attention mechanism. Here, we calculate multiple attention vectors $A$ from the LSTM hidden states, instead of just one single attention vector $a$. The attention matrix is then used to create multiple fixed-sized representations of the input sequence, with a different focus on the relevant parts of the sequences. The attention matrix is calculated as follows:

$$a = softmax\left(tanh\left(W_a h_t + b_a\right)V_a\right) \tag{4}$$

where $W_a$ and $W_b$ are weight matrices and $b_a$ is the bias of the attention function. The advantage of having multiple attention vectors is that some of them can be used to predict the position of the CS, as they are vectors of size equal to the sequence length $L$

summing to 1. Therefore, 4 of the 13 attention vectors that the model uses are used in the prediction of the *SP*, *mTP*, *cTP*, and *luTP* CS:

$$y' = f_{cs}(a^{1:4}) \tag{5}$$

To encode the sequence of hidden states $H = [h_1, ..., h_L]$ into a fixed sized matrix, the hidden states are multiplied by the attention matrix and summed up:

$$e = \sum_{t=1}^{L} a_t h_t \tag{6}$$

where $e$ matrix is the encoded representation of the protein sequence. $e$ holds a total of 13 different representations of the protein sequences; therefore, it is needed to summarise this matrix into a vector. This is done by a final feed-forward layer, which converts $E$ into a representation vector $e$. This is then used to calculate the output layer of the network, to predict the type of peptide (p) $y$

$$y = f_p(e) \tag{7}$$

Both outputs from the network $y$ and $y'$ are trained together. The exception is for proteins belonging to the negative set, that is, noTPs that lack a CS and, therefore, there is no error to back-propagate.

The model was trained and optimised using fivefold nested cross-validation. The four inner subsets were used to train the model, where three are used for training and one for validation to identify the best set of hyper-parameters. After optimisation, the fifth set, which was kept out of the optimisation, was used to assess the test set performance. This procedure was repeated using all five subsets as the test set. The advantage of this approach is that we obtain an unbiased test set performance on the whole dataset at the expense of having to train 5 × 4 = 20 models.

Different hyper-parameters were tested to find the best model such as the number of hidden units for the LSTM, attention and fully connected layers, number of attention vectors, the learning rate, and the dropout rate. We also experimented with a convolutional neural network as the initial layer, but the best results were achieved using a filter size of 1, which is equivalent to a fully connected layer along the feature dimension.

### Related tools

The tools included in the analysis adopt different machine learning algorithms intending to classify from one to many N-terminal sorting signals and the CS position. Most of the tools contain modules both for plant and nonplant proteins.

MitoFates (Fukasawa et al, 2015) combines amino acid composition and physicochemical properties with positively charged amphiphilicity, pre-sequence motifs, and position-weight matrices as input to a standard support vector machine classifier for modelling the mitochondrial pre-sequence and its CS.

PredSL (Petsalaki et al, 2006) uses neural networks, Markov chains, profile-hidden Markov models, and scoring matrices to classify proteins from the N-terminal amino acid sequence into five

**Table 1.** Performance of the predictors considering only the identification of the targeting peptides.

| Tool | Loc | Proteins | Precision | Recall | F1 score | MCC |
|------|-----|----------|-----------|--------|----------|-----|
| TargetP 2.0 | SP | 2697 | 0.97 | 0.98 | 0.98 | 0.97 |
| TargetP 1.1 | SP | 2697 | 0.86 | 0.97 | 0.91 | 0.89 |
| DeepLoc | SP | 2697 | 0.90 | 0.84 | 0.87 | 0.84 |
| PredSL | SP | 2697 | 0.69 | 0.90 | 0.78 | 0.73 |
| Predotar | SP | 2697 | 0.92 | 0.92 | 0.92 | 0.90 |
| MLP-20 | SP | 2697 | 0.93 | 0.93 | 0.93 | 0.91 |
| SignalP 5.0 | SP | 2697 | 0.99 | 0.99 | 0.99 | 0.98 |
| TargetP 2.0 | mTP | 499 | 0.87 | 0.85 | 0.86 | 0.86 |
| TargetP 1.1 | mTP | 499 | 0.32 | 0.90 | 0.48 | 0.51 |
| DeepLoc | mTP | 499 | 0.73 | 0.97 | 0.83 | 0.83 |
| PredSL | mTP | 499 | 0.18 | 0.93 | 0.31 | 0.37 |
| Predotar | mTP | 499 | 0.71 | 0.74 | 0.73 | 0.72 |
| TPpred 3 | mTP | 499 | 0.69 | 0.68 | 0.68 | 0.67 |
| MitoFates | mTP | 499 | 0.70 | 0.92 | 0.80 | 0.80 |
| MLP-20 | mTP | 499 | 0.69 | 0.58 | 0.63 | 0.62 |
| TargetP 2.0 | cTP | 227 | 0.90 | 0.86 | 0.88 | 0.88 |
| TargetP 1.1 | cTP | 227 | 0.39 | 0.88 | 0.54 | 0.58 |
| DeepLoc | cTP | 227 | 0.70 | 0.94 | 0.80 | 0.80 |
| PredSL | cTP | 227 | 0.16 | 0.78 | 0.27 | 0.34 |
| Predotar | cTP | 227 | 0.51 | 0.76 | 0.61 | 0.61 |
| TPpred 3 | cTP | 227 | 0.76 | 0.64 | 0.69 | 0.69 |
| MLP-20 | cTP | 227 | 0.51 | 0.37 | 0.43 | 0.40 |
| TargetP 2.0 | luTP | 45 | 0.75 | 0.75 | 0.75 | 0.75 |
| PredSL | luTP | 45 | 0.46 | 0.71 | 0.56 | 0.57 |
| MLP-20 | luTP | 45 | 0.10 | 0.02 | 0.04 | 0.05 |
| TargetP 2.0 | noTP | 9537 | 0.98 | 0.98 | 0.98 | 0.95 |
| TargetP 1.1 | noTP | 9537 | 0.99 | 0.84 | 0.91 | 0.75 |
| DeepLoc | noTP | 9537 | 0.95 | 0.95 | 0.95 | 0.83 |
| PredSL | noTP | 9537 | 0.99 | 0.60 | 0.75 | 0.52 |
| Predotar | noTP | 9537 | 0.96 | 0.95 | 0.95 | 0.84 |
| TPpred 3 | noTP | 9537 | 0.76 | 0.98 | 0.86 | 0.29 |
| MitoFates | noTP | 9537 | 0.75 | 0.98 | 0.85 | 0.25 |
| MLP-20 | noTP | 9537 | 0.95 | 0.97 | 0.96 | 0.85 |
| SignalP 5.0 | noTP | 9537 | 0.92 | 0.99 | 0.96 | 0.83 |

The table shows the performance in the test set yield by each predictor for mitochondria (*mTP*), chloroplast (*cTP*), thylakoid (*luTP*), *SP*, and other (*noTP*), in terms of F1 score, Matthews correlation coefficient (MCC), precision, and recall.

groups: chloroplast, thylakoid, mitochondrion, secretory pathway, and other.

SignalP 5.0 (Almagro Armenteros et al, 2019) is a deep neural network–based method combined with conditional random field that distinguishes between various types of SPs across all domains of life and between three kinds of prokaryotic SPs (Sec/SPI, Sec/SPII, and Tat/SPI).

TargetP 1.1 (Emanuelsson et al, 2000, 2007) classifies proteins into four different groups (SP, mTP, chloroplastic transit peptide, and other) using two layers of feed-forward neural networks and detects the CSs using a variety of methods, including position-weight matrices for the *mTPs*.

TPpred 3 (Savojardo et al, 2015) is a combination of a Grammatical Restrained Hidden Conditional Random Field, N-to-1 Extreme Learning Machines, and Support Vector Machines. It detects transit peptides, classifying them as mitochondrial or chloroplastic and localising their CSs.

For comparison, we also choose to include two methods that do identify the subcellular localisation of proteins but do not predict the CS of the targeting peptides.

Predotar (Small et al, 2004) is a three-layer feed-forward neural network-based approach to classify proteins in four different classes: SP, mTP, cTP, and other.

DeepLoc (Almagro Armenteros et al, 2017) uses a deep learning architecture very similar to what we have used in this study to predict the subcellular localisation of proteins.

MLP-XX is a simple multi-layer perceptron that we tested for comparison. MLP-XX consists of a one layer feed-forward neural network where using one hot encoding of the first XX amino acids as input (up to 20). It used the same cross-validation as TargetP 2.0. We examined the inclusion of different numbers of N-terminal residues, and the average F1 score increased from 0.77 when using five residues to 0.93 when using 20 (see Table 1). For comparison, we include MLP-20 in the results.

### Evaluation of the performance

We use several performance measures to obtain a uniform evaluation of the prediction. For the performance of sorting signals, we use the F1 score that may count as a harmonic average of the precision and recall. We also computed the Matthews Correlation Coefficient for each class to measure the ability to separate one type of target peptides from all other proteins (Baldi et al, 2000). In addition, we used precision and recall to examine the combined performance of sorting signals and CS. All these measurements were expressed in terms of "tp" = true positive, "tn" = true negative, "fp" = false positive, and "fn" = false negative.

$$precision = \frac{tp}{tp + fp} \tag{8}$$

$$recall = \frac{tp}{tp + fn} \tag{9}$$

$$F1 = 2\left(\frac{precision \times recall}{precision + recall}\right) \tag{10}$$

$$MCC = \frac{tp \times tn - fp \times fn}{\sqrt{(tp + fn)(tp + fp)(tn + fp)(tn + fn)}} \tag{11}$$

### Additional analysis

In several figures, standard or variations of sequence LOGOs are shown. These were generated using the Seq2Logo program (Thomsen & Nielsen, 2012). In addition to standard sequence LOGOs calculated from multiple sequence alignments, we generated two other types of LOGOs. First, LOGOs representing the frequency of amino acids in position 2 (and not the entropy) were generated to highlight differences in frequencies. Furthermore, LOGOs representing the strength of the attention layer output were also generated to analyse the importance of a position deduced by the machine learning method.

Secondary structure preferences for the different targeting peptides were calculated from the scale from (Delage & Roux, 1987).

The Log2 of the average preference was plotted for each residue in the different targeting peptides.

## Conclusions

Here, we introduce the new version of TargetP 2.0 that includes the prediction of thylakoid transit peptides and uses deep neural networks. TargetP 2.0 can be helpful to accurately annotate N-terminal sorting signals and CSs in particular as it scales to complete proteomes. TargetP 2.0 outperforms all other methods in all N-terminal sorting signals. Regarding classification, the only alternative method that comes close to TargetP 2.0 in performance is DeepLoc for SPs and mitochondrial transit peptides. However, for chloroplast peptides, TargetP 2.0 is superior, and DeepLoc does not predict thylakoid localisation. On the other hand, DeepLoc also predicts many other subcellular localisations not governed by targeting peptides.

When analysing how TargetP 2.0 arrives at its predictions, we note that two distinct regions contribute. As expected, the region around the CS is essential for classification of the type of transit peptide. However, surprisingly, an equally important contribution comes from the N-terminal region. Upon closer inspections, it is clear that (i) in plants, two-thirds of the chloroplast and luminal transit peptides have an alanine in position 2 (after the N-terminal methionine) and (ii) in fungi, only 20–30% of the N termini of *mTPs* and *SPs* can be cleaved, compared with 60% for noTPs. In summary, this indicates that it is not unlikely that specificity of MAPs aids in the co-translational targeting of peptides into organelles.

## Supplementary Information

## Acknowledgements

We thank Castrense Savojardo for having run TPpred 3 for our analysis. We thank Elzbieta Glaser for discussions about earlier work regarding the importance of alanine in position 2 in *cTPs*. We thank the NVIDIA Corporation and the Swedish National Infrastructure for Computing for providing computational resources. A Elofsson was supported by grant VR-NT-2016-03798 from the Swedish National Research Council.

### Author Contributions

JJA Armenteros: resources, software, formal analysis, investigation, methodology, and writing—original draft, review, and editing.
M Salvatore: resources, data curation, software, formal analysis, investigation, and writing—original draft, review, and editing.
O Emanuelsson: validation and writing—review and editing.
O Winther: conceptualization, resources, software, supervision, funding acquisition, methodology, and writing—review and editing.
G von Heijne: validation and writing—review and editing.

A Elofsson: data curation, formal analysis, supervision, funding acquisition, validation, investigation, visualization, and writing—original draft, review, and editing.
H Nielsen: resources, data curation, software, formal analysis, supervision, funding acquisition, validation, investigation, project administration, and writing—original draft, review, and editing.

**Conflict of Interest Statement**

The authors declare that they have no conflict of interest.

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
