## [Reviewer comments · Life Science Alliance]

Life Science Alliance

Detecting Sequence Signals in Targeting Peptides Using Deep Learning

Jose Almagro Armenteros, Marco Salvatore, Ole Winther, Olof Emanuelsson, Gunnar von Heijne, Arne Elofsson, and Henrik Nielsen

DOI: <https://doi.org/10.26508/lsa.201900429>

Corresponding author(s): Arne Elofsson, Stockholm University and Henrik Nielsen, DTU

Review Timeline:

Submission Date:	2019-05-15
Editorial Decision:	2019-07-05
Revision Received:	2019-09-05
Editorial Decision:	2019-09-17
Revision Received:	2019-09-18
Accepted:	2019-09-18

Scientific Editor: Andrea Leibfried

Transaction Report:

July 5, 2019

Re: Life Science Alliance manuscript #LSA-2019-00429

Prof. Arne Elofsson
Stockholm University
Biochemistry and Biophysics and Science for Life Laboratory
Box 1031
Solna 17121
Sweden

Dear Dr. Elofsson,

Thank you for submitting your manuscript entitled "Detecting Novel Sequence Signals in Targeting Peptides Using Deep Learning" to Life Science Alliance. The manuscript was assessed by expert reviewers, whose comments are appended to this letter. Please excuse the unusual delay in getting back to you. We were waiting for a third report on your work, which unfortunately was not delivered. We therefore decided to move forward now with the reports at hand.

As you will see, the reviewers think that the advance of TargetP 2.0 over other methods is small, but likely still of value to others. The reviewers raise, however, a few points that would need to get addressed. We would thus like to invite you to submit a revised version to us, addressing the individual criticisms raised. Importantly, the tool source code should be accessible (both reviewers). It would also be good to address the comment of rev#1 regarding the discrepancy found around the cleavage site of mitochondrial targeting peptides.

Thank you for this interesting contribution to Life Science Alliance. We are looking forward to receiving your revised manuscript.

Sincerely,

B. MANUSCRIPT ORGANIZATION AND FORMATTING:

Reviewer #1 (Comments to the Authors (Required)):

In this manuscript, the authors report an update of their renowned prediction tool, TargetP, using deep learning. Moreover, they claim that their inspection of its attention layer lead them to discover an interesting bias of amino acid frequency in several targeting peptides.

I agree that TargetP has been used quite extensively in wide communities and welcome its update very much. Indeed, its performance has been improved quite significantly, which will undoubtedly enhance its practical value. On the other hand, it relies on a method that is rather similar to the one used in DeepLoc, which shows almost comparable performance. As for their 'discoveries', their Fig. 1 (and Table S8 as well) is quite striking but, as they admit in P. 16, this may not be totally 'novel', contrary to their claim in the title. In addition, the sequence feature they found is quite simple and thus it is hard to believe that their machine learning approach has had some necessity for the discovery. Moreover, the discrepancy found around the cleavage site of mitochondrial targeting peptides shows that their current method is not ideal for detecting subtle sequence features. Therefore, I think that this manuscript might be appropriate to be published in journals more oriented to informatics domain.

Minor points:

1. In abstract, they give the URL of their tool but I think that it should be clearly shown in its main text, too. Similarly, for the sake of its reproducibility, all of the data used for its training/testing should be provided, at least (it is desirable, of course, that the authors would freely provide the source code of their tool, too).
2. The authors recently released a new version of SignalP. I wonder why they did not compare their performance in terms of the distinction between (what they call) SPs and noTPs. Probably, the meaning of using SignalP for eukaryotic proteins would have become diminished significantly.
3. In some figures, there exists position zero in amino acid sequences. I wonder how the numbering of sequences is defined.
4. I understand that the training has been done using all kingdoms of sequences. I know that deep learning requires a number of data but I still wonder if the prediction accuracy would show any improvement if the training is done using kingdom-specific data.
5. Again, I believe that the number of the data is too small but I think that they should mention about the possibility of predicting the bipartite mitochondrial targeting signals (for intermembrane space-located proteins) as well.
6. P. 13: for general readers, what is done in Ref. 29 should be briefly explained in this paper, too. In Table S7, maybe using the standard name for the organisms is better; the addition of percentages would be favorable.
7. Fig. S1: how were the few proteins chosen? The shaded areas were not clear in my environment. More importantly, isn't it possible to use the more detailed proteome data (such as Thul et al, Science 2017) for its verification?

Reviewer #2 (Comments to the Authors (Required)):

REVIEW of TargetP 2.0 MANUSCRIPT

DISCLAIMER/COI

I have worked with some of the authors in the past, and I am a co-author of one of the tools benchmarked in the manuscript.

SUMMARY

The authors extend their exceptionally strong record in applying machine learning classifiers to protein sorting signals.

They benchmark against several methods, including their own previous method and come up with a sizable improvement in prediction accuracy. The authors describe the particular architecture design choices (such as employment of LSTM) made in developing TargetP 2.0. It appears those choices were good ones and likely are responsible for the high benchmark performance of TargetP 2.0.

The authors also try to see if some biological insight can be obtained by analyzing the parameters learned by the neural network.

One observation they note is that there is a statistical correlation between the presence and type of transit peptide and the amino acid immediately following the initial methionine. They discuss the potential significance of this in terms of methionine aminopeptidase cleavage.

The paper is well written with effective figures and tables.

I expect TargetP 2.0 can be a valuable prediction tool, while the manuscript will be of interest to anyone developing classifiers using amino acid sequences as input.

PLEASE CONFIRM TOOL IS OPEN SOURCE

I have just one important thing to confirm:

(Probably I just missed it) but I did not see where the source code for the program can be downloaded.

This is of course essential for any academic work, so that the research community can examine, verify and extend the work.

COMMENT ON THE TITLE

A discretionary comment I have is that the title is perhaps a bit misleading.

Just looking at the title alone, 'Detecting Novel Sequence Signals' could mean simply prediction of instances of sequence signals not in the training data.

But judging from the abstract, I guess it is intended to give the reader the impression that the use of deep learning was somehow fundamental to a new discovery regarding sequence signals, in particular the alanine in position two.

However I think the identification of the second amino acid rather reflects the high quality of the researchers and their datasets. In the past (for example in my PhD thesis over 20 years ago) researchers have employed straightforward methods to systematically look for correlation between the sorting class of proteins and the amino acids at their start (and end); so this is not an idea which requires deep learning.

On the other hand, the authors have done excellent work and probably should be allowed to tell the story their way.

MINOR COMMENTS

Regarding typography, 'ITP' looked a bit like '1TP' in my pdf viewer. At the discretion of the authors, they may consider if a different font might produce a bigger vizual distinction between the intended 'ITP' and '1TP'.

> Predotar [10] is a three-layer feed-forward neural network-based approachable to approachable to...  approach to

> We also computed the Matthews Correlation Coefficient (MCC) for each class, to have a much more balanced evaluation of the prediction

It is not completely clear to me precisely what is being "balanced". The frequency of the classes??

> UniProt and TargetP 2.0 predictions is about 80% for the organelles and over 90% for signal peptides.

> The high agreement for SPs is quite likely due to UniProt applying SignalP [30] for its annotation of SPs

As a general comment it seems there may be some danger of circular logic in which prediction tools are used to annotate and then those annotations end up being used for benchmarking the performance of --- prediction tools.

Paul Horton
Department of Computer Science
National Cheng Kung University (NCKU)
Tainan, Taiwan

Reviewer #1 (Comments to the Authors (Required)):

In this manuscript, the authors report an update of their renowned prediction tool, TargetP, using deep learning. Moreover, they claim that their inspection of its attention layer lead them to discover an interesting bias of amino acid frequency in several targeting peptides.

We thank both the reviewers for their valuable comments. We have updated the manuscript and marked all changes in red.

I agree that TargetP has been used quite extensively in wide communities and welcome its update very much. Indeed, its performance has been improved quite significantly, which will undoubtedly enhance its practical value. On the other hand, it relies on a method that is rather similar to the one used in DeepLoc, which shows almost comparable performance. As for their 'discoveries', their Fig. 1 (and Table S8 as well) is quite striking but, as they admit in P. 16, this may not be totally 'novel', contrary to their claim in the title. In addition, the sequence feature they found is quite simple and thus it is hard to believe that their machine learning approach has had some necessity for the discovery.

We fully agree that the machine learning methodology was not necessary for the "discovery". However, it helped with the discovery. Further, although the distribution of amino acids in position two has been mentioned before, this is the first time the importance of this for predictions has been highlighted. Anyhow, we have changed the title of the paper to just: "Detecting Sequence Signals in Targeting Peptides Using Deep Learning" as well as highlighted that this is not entirely novel in the abstract. For more details, see comments to reviewer two.

Moreover, the discrepancy found around the cleavage site of mitochondrial targeting peptides shows that their current method is not ideal for detecting subtle sequence features. Therefore, I think that this manuscript might be appropriate to be published in journals more oriented to informatics domain.

Although our predictions of cleavage peptides are not perfect, we do believe that it is a significant step forward compared with earlier methods. Anyhow, we have tried to clarify why the attention layers focus on a single position in the text.

Minor points:

1. In abstract, they give the URL of their tool but I think that it should be clearly shown in its main text, too. Similarly, for the sake of its reproducibility, all of the data used for its training/testing should be provided, at least (it is desirable, of course, that the authors would freely provide the source code of their tool, too).

All data and the source code (licensed under "CC BY-NC-SA") is now available from the website. We have also added the URL to the main text.

2. The authors recently released a new version of SignalP. I wonder why they did not compare their performance in terms of the distinction between (what they call) SPs and noTPs. Probably, the meaning of using SignalP for eukaryotic proteins would have become diminished significantly.

We have added the comparison with SignalP 5.0. As expected the performance is similar - but SignalP 5.0 is better at predicting the cleavage site. This might be due (i) the use of a conditional random field to identify the cleavage site or (ii) or because many if the proteins we tested are included in the training set of SignalP 5.0. It can also be noted that eukaryotic group specific versions of SignalP 4.0 was tested but they did not show any improvement over a general model.

3. In some figures, there exists position zero in amino acid sequences. I wonder how the numbering of sequences is defined.

Thanks for detecting this. The numbering of protein sequences is always a bit inconsistent. Now we have tried to use position 1 for the N-terminal (methionine) position in the sequence consistently. In addition, we have skipped the "zero" when discussing the cleavage site, so that position -1 is just before the cleavage and +1 is the next position. Although this is illogical - we decided to keep this annotation as it has become the standard in the field.

4. I understand that the training has been done using all kingdoms of sequences. I know that deep learning requires a number of data but I still wonder if the prediction accuracy would show any improvement if the training is done using kingdom-specific data.

We examined the possibility to train kingdom-specific versions of TargetP2.0 - unfortunately, this resulted in a decrease in performance by about 5%. Most likely this is due to smaller training sets and that the Target peptides do not differ

significantly between the kingdoms.

5. Again, I believe that the number of the data is too small but I think that they should mention about the possibility of predicting the bipartite mitochondrial targeting signals (for intermembrane space-located proteins) as well.

We thank the reviewer for this suggestion. Unfortunately, the number of annotated intermembrane space-located proteins with transit peptides in uniprot is small (118 in total and only 37 with any experimental annotation and none of these has annotation about a bipartite signal). Therefore, we do not think it is possible to predict bipartite mitochondrial peptides with our approach.

6. P. 13: for general readers, what is done in Ref. 29 should be briefly explained in this paper, too. In Table S7, maybe using the standard name for the organisms is better; the addition of percentages would be favorable.

We changed the names and added percentages in Table S7. We also added a description on how to compensate for errors in large studies as proposed by Marot-Lassauzaie, Bernhofer, and Rost; this reads as follows:

In this method, the predicted number of proteins in a class is calculated by using the number of members predicted to that class and the estimated number of mispredictions from other classes. First, the fraction of misprediction between classes is calculated. The estimated number of members of one class is then calculated from the predicted number in that class, deducing the estimated fraction of false predictions. Thereafter, the estimated number of mispredictions from other classes is added from the number of predicted members in a class multiplied with the estimated mispredictions from that class to the first class.

7. Fig. S1: how were the few proteins chosen? The shaded areas were not clear in my environment. More importantly, isn't it possible to use the more detailed proteome data (such as Thul et al, Science 2017) for its verification?

The proteins were chosen randomly and are only here for illustration (basically all proteins look similar). For complete (and statistical) analysis the LOGOs (Fig 5,6, S2) are much more illustrative. We have clarified this in the text.

Reviewer #2 (Comments to the Authors (Required)):

REVIEW of TargetP 2.0 MANUSCRIPT

DISCLAIMER/COI

I have worked with some of the authors in the past, and I am a co-author of one of the tools benchmarked in the manuscript.

We thank both the reviewers for their valuable comments. We have updated the manuscript and marked all changes in red.

SUMMARY

The authors extend their exceptionally strong record in applying machine learning classifiers to protein sorting signals. They benchmark against several methods, including their own previous method and come up with a sizable improvement in prediction accuracy. The authors describe the particular architecture design choices (such as employment of LSTM) made in developing TargetP 2.0. It appears those choices were good ones and likely are responsible for the high benchmark performance of TargetP 2.0.

The authors also try to see if some biological insight can be obtained by analyzing the parameters learned by the neural network. One observation they note is that there is a statistical correlation between the presence and type of transit peptide and the amino acid immediately following the initial methionine. They discuss the potential significance of this in terms of methionine aminopeptidase cleavage.

The paper is well written with effective figures and tables. I expect TargetP 2.0 can be a valuable prediction tool, while the manuscript will be of interest to anyone developing classifiers using amino acid sequences as input.

Thanks.

PLEASE CONFIRM TOOL IS OPEN SOURCE. I have just one important thing to confirm: (Probably I just missed it) but I did not see where the source code for the program can be downloaded. This is of course essential for any academic work, so that the research community can examine, verify and extend the work.

We have released TargetP2.0 under the creative commons "CC BY-NC-SA" that

allows non-commercial use of the program. The source code is now available

from: <http://github.com/JJAlmagro/TargetP-2.0>

COMMENT ON THE TITLE: A discretionary comment I have is that the title is perhaps a bit misleading. Just looking at the title alone, 'Detecting Novel Sequence Signals' could mean simply prediction of instances of sequence signals not in the training data. But judging from the abstract, I guess it is intended to give the reader the impression that the use of deep learning was somehow fundamental to a

new discovery regarding sequence signals, in particular the alanine in position two.

We changed the title to "Detecting Sequence Signals in Targeting Peptides Using Deep Learning"

However I think the identification of the second amino acid rather reflects the high quality of the researchers and their datasets. In the past (for example in my PhD thesis over 20 years ago) researchers have employed straightfoward methods to systematically look for correlation between the sorting class of proteins and the amino acids at their start (and end); so this is not an idea which requires deep learning.

We agree that deep learning is not at all fundamental for the discovery of the sequence signal in position two. It was just (largely) overlooked by the bioinformatics community (including ourselves) for the last twenty years. Anyhow, we agree that the title might be misleading and therefore we changed it (but not the story). Also, see comments to reviewer one. We added some clarification in the abstract also.

On the other hand, the authors have done excellent work and probably should be allowed to tell the story their way.

Thanks.

MINOR COMMENTS

Regarding typography, 'lTP' looked a bit like '1TP' in my pdf viewer. At the discretion of the authors, they may consider if a different font might produce a bigger vizual distinction between the intended 'lTP' and '1TP'.

We have changed all occurrences of lTP to luTP. We also changed all (TPs) to italics. Hopefully, this helps readability.

"Predotar [10] is a three-layer feed-forward neural network-based approachable to approachable to..."  approach to

Changed

"We also computed the Matthews Correlation Coefficient (MCC) for each class, to have a much more balanced evaluation of the prediction." It is not completely clear to me precisely what is being "balanced". The frequency of the classes??

We agree that this is unclear. What we meant was that MCC is a better measure than fraction correct if there is one class that is much larger than the other class. We reformulated it. Hopefully, it is clearer now.

"UniProt and TargetP 2.0 predictions is about 80% for the organelles

and over 90% for signal peptides. The high agreement for SPs is quite likely due to UniProt applying SignalP [30] for its annotation of SPs" As a general comment it seems there may be some danger of circular logic in which prediction tools are used to annotate and then those annotations end up being used for benchmarking the performance of --- prediction tools.

We fully agree. First, we have to highlight that we do only use experimentally verified annotations for training. The comparison with uniprot is not done for evaluation of the performance but to examine what happens with large scale analysis of current tools. In particular for the groups with only few experimentally verified peptides. Anyhow, we do think it has some value to make this comparison as this tells something about how a prediction method can be applied on a larger scale. However, we have tried to clarify that (i) we do not believe uniprot is the truth and (ii) we should be careful with this type of comparisons.

September 17, 2019

RE: Life Science Alliance Manuscript #LSA-2019-00429R

Prof. Arne Elofsson
Stockholm University
Biochemistry and Biophysics and Science for Life Laboratory
Box 1031
Solna 17121
Sweden

Dear Dr. Elofsson,

Thank you for submitting your revised manuscript entitled "Detecting Sequence Signals in Targeting Peptides Using Deep Learning". As you will see, original reviewer #1 re-assessed your manuscript and appreciates the introduced changes and we would thus be happy to publish your paper in Life Science Alliance pending final revisions necessary to meet our formatting guidelines:

- please address the remaining comment of reviewer #1
- please upload your manuscript text in word docx format
- please upload all figures as individual files (without legends)

A. FINAL FILES:

-- Summary blurb (enter in submission system): A short text summarizing in a single sentence the study (max. 200 characters including spaces). This text is used in conjunction with the titles of papers, hence should be informative and complementary to the title. It should describe the context

and significance of the findings for a general readership; it should be written in the present tense and refer to the work in the third person. Author names should not be mentioned.

B. MANUSCRIPT ORGANIZATION AND FORMATTING:

Sincerely,

Andrea Leibfried, PhD
Executive Editor
Life Science Alliance
Meyerohofstr. 1
69117 Heidelberg, Germany
t +49 6221 8891 502
e a.leibfried@life-science-alliance.org
www.life-science-alliance.org

Reviewer #1 (Comments to the Authors (Required)):

In this revised manuscript, I think that the authors have faithfully addressed points raised by the reviewers, including myself. I think that the manuscript is now acceptable, though i feel that the scientific names in Table S7 are better to be shown in italics.

September 18, 2019

RE: Life Science Alliance Manuscript #LSA-2019-00429RR

Prof. Arne Elofsson
Stockholm University
Biochemistry and Biophysics and Science for Life Laboratory
Box 1031
Solna 17121
Sweden

Dear Dr. Elofsson,

Thank you for submitting your Methods entitled "Detecting Sequence Signals in Targeting Peptides Using Deep Learning". It is a pleasure to let you know that your manuscript is now accepted for publication in Life Science Alliance. Congratulations on this interesting work.

DISTRIBUTION OF MATERIALS:

Again, congratulations on a very nice paper. I hope you found the review process to be constructive and are pleased with how the manuscript was handled editorially. We look forward to future exciting submissions from your lab.

Sincerely,
